# Effects of Pneumatic Compression and Manual Massage on Recovery and Performance in Elite Brazilian Under-20 Soccer Players: A Crossover Trial

**DOI:** 10.3390/sports13090304

**Published:** 2025-09-03

**Authors:** Tiago Costa Esteves, Júlio Cesar de Oliveira Muniz Cunha, Júlio Guilherme Silva, María Rúa-Alonso, Luciano Teixeira dos Santos, Laercio Brehner Gemaque do Couto, José Vilaça-Alves, Estêvão Rios Monteiro, Igor Ramathur Telles de Jesus

**Affiliations:** 1Graduate Program in Rehabilitation Science, Centro Universitário Augusto Motta (PPGCR, UNISUAM), Rio de Janeiro 21031-060, Brazil; tiagoesteves@souunisuam.com.br (T.C.E.); cinesioliveira2@gmail.com (J.C.d.O.M.C.);; 2Undergraduate Program in Physical Therapy, Faculdade Inspirar, Belém 66055-080, Brazil; 3Graduate Program in Rehabilitation Science, Universidade Federal do Rio de Janeiro (PPGCR, UFRJ), Rio de Janeiro 21.941-902, Brazil; jglsilva@yahoo.com.br; 4Performance and Health Group, Department of Physical Education and Sport, Faculty of Sports Sciences and Physical Education, University of A Coruna, 15179 A Coruña, Spain; maria.rua@udc.es; 5School of Physical Education, Physiotherapy and Dance, Federal University of Rio Grande do Sul (UFRGS), Porto Alegre 90690-200, Brazil; 6School of Physical Education and Physiotherapy, Federal University of Pelotas (UFPel), Pelotas 96055-630, Brazil; 7Department of Sports Sciences, Exercise and Health, Universidade de Trás-os-Montes e Alto Douro, 5000-801 Vila Real, Portugal; 8Research Center in Sports Sciences, Health Sciences and Human Development (CIDESD), 5001-801 Vila Real, Portugal; 9Graduate Program in Biopsychosocial Health, Centro Universitário Augusto Motta (UNISUAM), Rio de Janeiro 21041-020, Brazil

**Keywords:** muscle fatigue, functional recovery, creatine kinase, physical performance, football

## Abstract

**Introduction**: Acute neuromuscular fatigue impairs athletic performance and increases the risk of musculoskeletal injury. Recovery strategies such as manual massage (MM) and intermittent pneumatic compression (IPC) have been proposed to mitigate these effects, although their efficacy in elite youth soccer remains under debate. **Objective**: To compare the acute effects of MM and IPC on muscle damage recovery, lower limb strength, and power in Brazilian Under-20 soccer athletes. **Methods**: A randomized crossover study was conducted with twenty male youth athletes (18.65 ± 0.67 years) from the under-20 category of Paysandu Sport Club—Brazil. Each athlete underwent both MM and IPC interventions, separated by a seven-day washout. Variables assessed included serum creatine kinase (CK), quadricep and hamstring isometric voluntary contraction (IVC), and vertical jump (VJ). **Results**: MM resulted in a CK reduction of Δ = −77.1 U/L (*p* = 0.042; d = 0.37), indicating a moderate effect size, while IPC induced a larger reduction of Δ = −138.0 U/L (*p* = 0.160; d = 1.41), with a very large effect size despite the lack of statistical significance. Neither intervention produced significant changes in quadricep or hamstring IVC, nor in VJ height (*p* > 0.05). **Conclusions**: Both MM (statistical difference) and IPC (clinical difference) were viable recovery strategies for attenuating acute serum CK without impairing neuromuscular performance in elite under-20 soccer players.

## 1. Introduction

Acute neuromuscular fatigue is a common physiological response in high-demand intermittent sports such as soccer and can significantly impair athletic performance while increasing the risk of musculoskeletal injuries [1,2]. This type of fatigue negatively affects muscle contractile capacity, strength, and motor coordination and may persist for up to 72 h post-exercise [3,4]. As a result, post-exercise recovery strategies have become increasingly adopted to mitigate the deleterious effects of repeated exertion and optimize the return to functional condition [5].

Among these strategies, manual massage (MM) and intermittent pneumatic compression (IPC) stand out and are widely used in athletic settings. MM is recognized for its analgesic effects, its ability to reduce muscle stiffness, and its enhancement of perceived recovery [1]. However, the evidence regarding its effects on muscular strength and power remains inconclusive [2,3]. In contrast, IPC has been employed to enhance venous and lymphatic return and to facilitate the clearance of pro-inflammatory metabolites, potentially reducing delayed-onset muscle soreness and serum creatine kinase (CK) levels [4]. Nevertheless, controlled trials suggest that the effects of IPC on neuromuscular performance remain controversial, particularly in acute settings [5,6].

Recent studies have indicated that massage tends to reduce CK levels in physically active populations [2], while IPC may present variable outcomes depending on the timing and mode of application [7]. Furthermore, reference values among elite youth soccer players suggest that baseline CK levels may range from 200 to 400 U/L, quadricep isometric strength (Q-IVC) typically ranges from 40 to 50 kgf, and vertical jump (VJ) performance ranges from 41.6 to 65 cm [8]. These reference points offer context for interpreting recovery outcomes in this population.

Although MM and IPC are widely used, direct comparisons between both interventions in youth athletic populations remain scarce. A structured search using PubMed, Scopus, and Web of Science with the terms “manual massage”, “intermittent pneumatic compression”, “recovery”, and “youth athletes” revealed a predominance of studies addressing each intervention separately, with few protocols applying both strategies under the same controlled design and virtually none focusing on elite youth soccer players. Moreover, research concurrently evaluating biochemical markers (CK) and neuromuscular variables (IVC, VJ) in crossover trials remains particularly limited.

Thus, this study aimed to compare the acute effects of MM and IPC on muscle damage recovery, lower limb strength, and power in Brazilian Under-20 soccer athletes.

## 2. Materials and Methods

### 2.1. Study Design

A randomized, controlled, crossover trial with a within-subjects design was conducted to compare the effects of manual massage (MM) and intermittent pneumatic compression (IPC) on recovery and performance. All participants received both interventions in a randomized order, with a seven-day washout period between sessions. All procedures were conducted in accordance with Law 14.874/2024 of the Brazilian National Health Council. The study was submitted and approved by the Augusto Motta University Centre ethics committee [7.163.787], and it was conducted in accordance with the Declaration of Helsinki.

### 2.2. Participants

Twenty male youth athletes (Table 1) from the under-20 category of Paysandu Sport Club—Brazil were recruited. All were regularly engaged in the club’s structured soccer training program and had a minimum of two years of competitive experience—an inclusion criterion adopted to ensure homogeneity in neuromuscular adaptation and familiarity with post-exercise recovery procedures. Participants were considered clinically healthy if they presented no diagnosed medical conditions or musculoskeletal injuries in the three months preceding the study, as confirmed by the club’s standard preseason clinical examination and laboratory screening. Additional inclusion criteria were (i) absence of musculoskeletal injuries within the last six months and (ii) regular participation in structured soccer training programs (at least three times per week). Exclusion criteria comprised the use of anti-inflammatory medications; the presence of neurological, cardiovascular, or musculoskeletal disorders; and any medical contraindication to the recovery techniques under investigation.

### 2.3. Procedures

All procedures (Figure 1) were randomized (by computer software—www.randomizer.org) and counterbalanced across subjects and experimental conditions to compare the MM and IPC effects on muscle damage recovery, lower limb strength, and power in under-20 soccer athletes from Paysandu Sport Club—Brazil. Randomization and allocation concealment were ensured, as sequence generation was performed by an independent researcher not involved in data collection or analysis. Although participant blinding was not feasible due to the tactile nature of the interventions, outcome assessors remained blinded to minimize measurement bias. A seven-day washout period was implemented between visits to prevent carry-over effects. This duration is supported by evidence indicating that serum CK levels typically peak within 24 h post-exercise and return to baseline by 48–72 h in trained individuals [9]. Subjects visited the laboratory twice, separated by this interval. During the IPC session, participants received 20 min of bilateral lower limb pneumatic compression. The MM condition was performed by a trained physiotherapists on both limbs, unilaterally, for 2.5 min on each treated region (e.g., anterior and posterior thigh and posterior lower leg). The order of region and limb was determined using a Latin square design to minimize expectancy effects regarding treatment location. This approach ensured equalization of total treatment time, 20 min for both IPC and MM conditions, and preserved single-blind allocation regarding the treated region during MM. A crossover design increased statistical power and reduced inter-individual variability.

### 2.4. Instruments

#### 2.4.1. Intermittent Pneumatic Compression (IPC)

The IPC intervention was performed using a Reboot Go^®^ (Recovery Reboot Sports Model, São Paulo, Brazil) sequential compression device. The equipment was applied to the lower limbs via inflatable compression boots with serially arranged chambers that promoted intermittent compression in a distal-to-proximal sequence. The protocol consisted of a compression cycle at a fixed pressure of 80 mmHg, applied continuously for a total of 20 min. This configuration was based on parameters supported by Maia et al. [10], who identified this pressure range as commonly adopted in sports recovery protocols and associated with reductions in muscle soreness, fatigue perception, and serum creatine kinase levels. The setup was standardized for all participants to ensure consistency and reproducibility of mechanical stimuli, aiming to enhance venous return and facilitate physiological recovery following physical exertion.

#### 2.4.2. Manual Massage (MM)

The MM intervention followed an adapted protocol from Abrantes et al. [11], totaling 20 min of application. All massage techniques were administered unilaterally by a single trained physiotherapist, each with an average of seven years of clinical experience in manual therapy and sports rehabilitation, ensuring bilateral application, procedural consistency, and reproducibility. Participants were first placed in the supine position, and effleurage and petrissage techniques were applied bilaterally to the quadriceps musculature, from distal to proximal, beginning at the mid-thigh and extending to the inguinal region, for 2.5 min per limb. They were then repositioned in the prone position, and the same techniques were applied unilaterally to the hamstrings (from the popliteal fossa to the ischial tuberosity) and gastrocnemius muscles (from the Achilles tendon to the gastrocnemius heads at the popliteal region), also for 2.5 min per limb. Pressure intensity was self-regulated by participants using a 10-point Visual Analog Scale (VAS), aiming to maintain a discomfort level between 4 and 5, consistent with validated methods for modulating massage intensity in sports contexts [12].

### 2.5. Measurements

#### 2.5.1. Serum Creatine Kinase (CK) Levels

Muscle fatigue was assessed through serum CK analysis using a portable biochemical analyzer Simplex ECO POC^®,^ (Corinto, MG, Brazil). Blood samples were collected at two time points: pre-intervention (CK-pre) and five minutes post-intervention (CK-post). This early post-intervention measurement was intentionally selected to capture the immediate acute response to each recovery strategy, consistent with the study’s primary objective and the short recovery windows typical of elite competitive environments. Similar early sampling protocols have been reported in sports recovery studies [13]. Samples (30 μL) were obtained via capillary puncture from the index finger, following antisepsis with 70% alcohol, with the first drop discarded. Reagents were prepared at room temperature, and cartridges were discarded after each reading, in accordance with the standardized procedures described by Leite et al. [13], ensuring methodological consistency and minimizing bias. The Simplex ECO POC^®^ employs disposable reagent cartridges and photometric detection, delivering quantitative CK measurements with a coefficient of variation below 5%, supporting the reliability and reproducibility of the biochemical assessment.

#### 2.5.2. Vertical Jump Performance (VJ)

VJ performance was evaluated using the Jumptest^®^ contact platform (Hidrofit Ltda, MG, Brazil), connected to the Multisprint^®^ software 1.1, which has been validated by Ferreira et al. [14]. This equipment has demonstrated high reliability for measuring countermovement jump height. After familiarization with the equipment, each athlete performed two submaximal jumps, followed by three maximal-effort vertical jumps, with a 10 s rest interval between attempts. Data were automatically processed by the software system.

#### 2.5.3. Isometric Voluntary Contraction (IVC)

IVC strength of the knee extensor (Q-IVC) and flexor (H-IVC) muscle groups was measured using the SP Tech^®^ portable isometric dynamometer (Manufactured in SC, Brazil), which has demonstrated excellent test–retest reliability in assessments of lower limb muscles, following the protocol described by Pinto-Ramos et al. [15]. For Q-IVC, participants were positioned in a seated posture with the knee flexed at 60° and stabilized using straps at the hip joint and mid-thigh. The dynamometer was affixed 5 cm proximal to the medial malleolus on the anterior aspect of the leg. For H-IVC, participants were positioned prone, with the knee flexed at 30°, stabilized at the lumbar spine and thigh. The dynamometer was positioned on the posterior aspect of the leg, also 5 cm proximal to the medial malleolus.

### 2.6. Statistical Analysis

Descriptive statistics are presented as means and standard deviations (SDs) for all outcome variables. Assumptions of normality were assessed using the Shapiro–Wilk test, supported by visual inspection of histograms and Q–Q plots, as well as evaluation of skewness and kurtosis. Homogeneity of variances was evaluated with Levene’s test. When the assumption of sphericity was violated (as tested by Mauchly’s test), Greenhouse–Geisser corrections were applied.

Given the within-subjects crossover design, two-way repeated measures analyses of variance (ANOVAs) were employed to examine the main effects of intervention (MM vs. IPC), time (pre- and post-intervention), and their interaction on the dependent variables: Q-IVC, H-IVC, VJ, and serum CK levels. Post hoc pairwise comparisons were adjusted using Bonferroni correction to control for Type I error inflation. In addition to *p*-values, partial eta squared (ηp^2^) was reported as a measure of effect size for ANOVA main effects and interactions, while mean difference (Δ) with 95% confidence intervals (95% CIs) and effect size (Cohen’s d) were calculated for each variable to improve the interpretation of the results and assess potential clinical relevance. This approach provides a more comprehensive view of the data, reducing reliance on significance testing alone and allowing for identification of potentially meaningful effects that may not reach statistical significance. All data were organized using Microsoft Excel 360^®^, and statistical analyses were performed in R (version 4.5) via the RStudio 4.5 interface. The level of significance was set at *p* < 0.05.

## 3. Results

Table 2 displays the outcomes following MM, and Table 3 presents the outcomes following IPC, both in terms of biochemical and neuromuscular parameters before and after the interventions.

As shown in Table 4, the repeated-measures ANOVA revealed no main effect of intervention for any outcome (all *p* > 0.19). A significant main effect of time was observed only for CK (F_(1,19)_ = 7.168, *p* = 0.015, ηp^2^ = 0.274), indicating an overall reduction from pre-intervention to post-intervention. However, the intervention × time interaction was not significant for CK, Q-IVC, H-IVC, or VJ (all *p* > 0.07), demonstrating that the pre-to-post changes did not differ significantly between MM and IPC.

### 3.1. Serum CK Concentration

A significant reduction in CK levels was observed following the MM intervention, whereas the IPC condition showed a greater, though non-significant, decrease accompanied by a large effect size (d = 1.41). This magnitude suggests a potentially meaningful clinical benefit, and the lack of statistical significance may be attributed to sample size limitations and inter-individual variability rather than an absence of physiological effect (see Table 2 and Table 3).

### 3.2. Isometric Voluntary Contraction (IVC)

Neither intervention produced significant changes in isometric voluntary contractions for either the quadriceps or hamstrings. While a slight reduction in Q-IVC was observed in both protocols, H-IVC showed a modest increase. These results indicate that the applied recovery techniques had no meaningful impact on isometric muscle strength (see Table 2 and Table 3).

### 3.3. Vertical Jump Performance

Vertical jump height remained unaffected by either recovery method. Minor declines were seen in both conditions, with no statistical or clinical relevance, suggesting stability in explosive neuromuscular performance post-intervention (see Table 2 and Table 3).

## 4. Discussion

Given the exploratory nature of the present study and the absence of prior research using comparable methodology in elite under-20 soccer players, our aim was to investigate potential differences in the acute recovery effects of MM and IPC. The findings revealed a very large effect size for IPC (d = 1.41) and a moderate effect size for MM (d = 0.37) in CK reduction, despite the lack of statistical significance for IPC. These differences may be attributed to the distinct physiological mechanisms of each intervention: MM has been reported to modulate soreness perception and reduce muscle stress markers through mechanical stimulation of tissues [11,16,17,18], whereas IPC may enhance venous return, lymphatic drainage, and metabolite clearance more rapidly in the acute phase [13]. From a theoretical standpoint, these results suggest that the long-standing assumption of MM superiority, common in parts of the massage literature [18], may not hold in this specific athletic context. Instead, both interventions appear to be viable recovery strategies, with IPC showing potential for greater biochemical impact, warranting further investigation with larger samples. These findings contribute to refining the theoretical framework for recovery strategies by emphasizing the need to integrate both statistical and clinical relevance when interpreting results in sports performance research [16,17].

These findings align with the revised exploratory hypothesis, which sought to investigate potential differences in the acute recovery effects of MM and IPC without presuming superiority of one intervention. MM promoted a statistically significant reduction in CK, consistent with its reported capacity to modulate muscle stress markers. IPC also induced CK reduction, with a very large effect size (d = 1.41), suggesting a potentially meaningful physiological impact despite the lack of statistical significance, likely influenced by sample size and inter-individual variability. This observation aligns with previous studies in the literature, in which IPC effects on CK and muscle function have been inconsistent in statistical terms but often show favorable trends in perceived recovery and inflammatory marker reduction [10,19]. The absence of statistically significant improvements in Q-IVC, H-IVC, or VJ following either intervention is consistent with existing evidence. Hilbert et al. [20] and Dawson et al. [21] found no post-massage strength improvements, while Davis et al. [22], in a meta-analysis of 29 trials, reported no enhancements in strength, power, speed, or endurance. These findings may reflect the acute nature of the interventions, the limited sensitivity of some outcome measures, and the athletes’ high baseline conditioning. From a physiological perspective, both MM and IPC act through passive mechanisms: MM stimulates cutaneous and deep mechanoreceptors, promoting parasympathetic modulation and reduced nociceptive input [6], whereas IPC mimics the skeletal muscle pump, facilitating venous return and lymphatic drainage [23]. Neither modality, however, elicits neuromuscular activation sufficient to generate immediate strength or power gains. The current results reinforce their role in preserving performance rather than enhancing it. In elite competitive contexts, the ability to maintain neuromuscular function without decline is itself a valuable outcome, supporting the use of both MM and IPC as effective recovery strategies in the short term.

This study presents inherent limitations that must be acknowledged. First, although the sample included the full roster of eligible under-20 athletes from a professional soccer team, the relatively small sample size may have limited the statistical power to detect smaller but potentially meaningful effects, particularly in functional performance variables. This raises the possibility of Type II error, especially in the context of high inter-individual variability common in athletic populations. Additionally, while the crossover design improves internal validity and reduces inter-subject variability, its structure assumes a stable baseline and full recovery between phases, which, despite the washout period, cannot be guaranteed without more extended longitudinal monitoring. Importantly, the exploratory hypothesis initially aimed to investigate potential differences between MM and IPC without presuming superiority. The results indicated a larger effect size for IPC in CK reduction, albeit without statistical significance, which differs from the prevailing assumption in parts of the massage literature. This mismatch highlights the need for cautious interpretation and suggests that the relative efficacy of MM and IPC may depend on specific recovery outcomes and assessment time frames. These limitations restrict the generalizability of the findings and underscore the need for replication in larger, multi-center trials with extended follow-up to better elucidate the clinical relevance of these effects.

## 5. Conclusions

Both MM and IPC were evaluated for their acute effects on muscle damage and neuromuscular performance in elite under-20 soccer players. MM produced a statistically significant reduction in serum CK levels, with a moderate effect size, indicating consistent benefits in post-exercise physiological recovery. IPC also reduced serum CK, but without statistical difference, demonstrating a large effect size, which may reflect the limited sample size and inter-individual variability. Neither intervention resulted in significant changes in isometric strength or vertical jump performance, suggesting that both strategies effectively preserved neuromuscular function in the short term. These findings support the use of both MM and IPC as viable recovery options in high-demand athletic environments, with IPC warranting further investigation for its potential biochemical impact. Future research should examine the effects of repeated sessions, explore individual response variability, and assess performance outcomes over longer recovery periods to optimize the application of these interventions.

## Figures and Tables

**Figure 1 sports-13-00304-f001:**
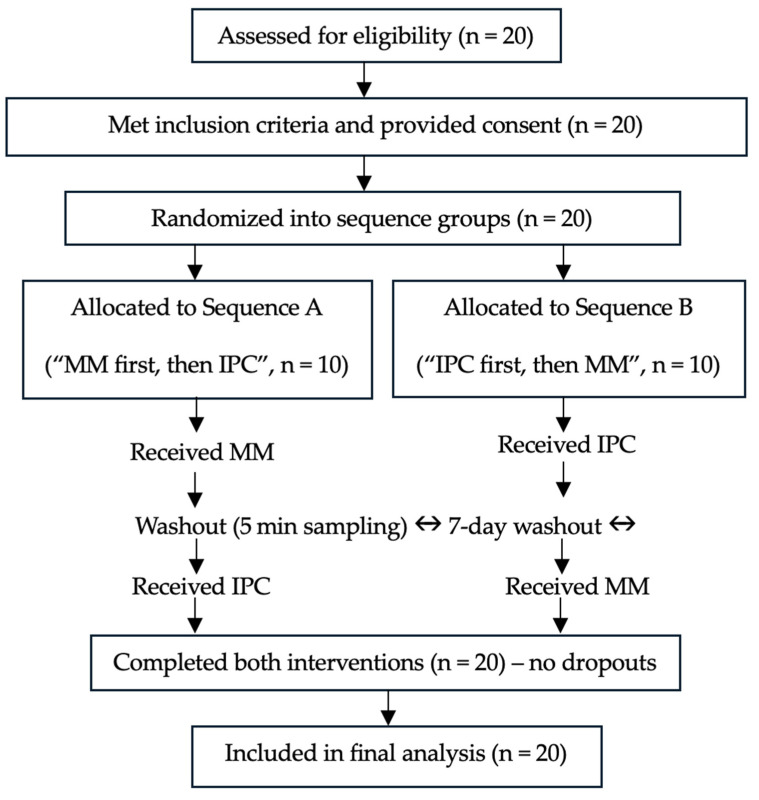
Participant flow diagram based on CONSORT guidelines adapted for crossover trials. All 20 participants completed both intervention phases (manual massage [MM] and intermittent pneumatic compression [IPC]) without exclusions or dropouts. Randomization defined the order of treatments with a 7-day washout period between sessions.

**Table 1 sports-13-00304-t001:** Anthropometric characteristics of the sample (Mean ± SD and 95% CI).

	Mean ± SD	95% CI	CV (%)
Age (years)	18.65 ± 0.67	18.40–18.90	3.59
Height (m)	1.78 ± 0.07	1.75–1.81	3.93
Weight (kg)	72.8 ± 6.99	70.19–75.41	9.60
Body Fat (%)	12.3 ± 1.63	11.69–12.91	13.25

SD: standard deviation; 95%CI: 95% confidence interval; CV: coefficient of variation.

**Table 2 sports-13-00304-t002:** Pre- and post-intervention values (mean, standard deviation, 95% confidence interval, *p*-value, and effect size) for serum CK, isometric strength, and vertical jump performance following manual massage.

Variable	Pre (Mean ± SD)	Post (Mean ± SD)	Δ Mean (95% CI)	*p*-Value	Cohen’s d
CK (U/L)	382.50 ± 163.00	305.40 ± 240.00	−77.10 (−151.3 to −2.9)	0.042 *	0.37
QIVC (kgf)	44.64 ± 12.39	42.74 ± 9.56	−1.90 (−8.9 to 5.1)	0.590	0.17
HIVC (kgf)	29.27 ± 8.16	29.84 ± 7.14	+0.57 (−2.9 to 4.0)	0.740	0.07
VJ (cm)	39.81 ± 3.40	39.29 ± 3.34	−0.52 (−2.3 to 1.3)	0.530	0.15

* statistical difference between post–pre comparison. Mean and standard deviation (SD) values pre- and post-intervention for manual massage (MM), along with mean difference (Δ) and its 95% confidence interval (95% CI), *p*-value (paired Student’s *t*-test), and effect size (Cohen’s d). Effect sizes were interpreted as follows: small (≥0.2), medium (≥0.5), large (≥0.8), and very large (>1.2).

**Table 3 sports-13-00304-t003:** Pre- and post-intervention values (mean, standard deviation, 95% confidence interval, *p*-value, and effect size) for serum CK, isometric strength, and vertical jump performance following intermittent pneumatic compression.

Variable	Pre (Mean ± SD)	Post (Mean ± SD)	Δ Mean (95% CI)	*p*-Value	Cohen’s d
CK (U/L)	337.00 ± 114.50	199.00 ± 89.50	−138.00 (−341.7 to 65.7)	0.160	1.41
QIVC (kgf)	45.08 ± 12.92	43.07 ± 10.18	−2.01 (−9.6 to 5.6)	0.580	0.18
HIVC (kgf)	28.91 ± 8.66	30.14 ± 7.96	+1.23 (−2.2 to 4.7)	0.470	0.15
VJ (cm)	39.58 ± 4.09	39.23 ± 3.41	−0.35 (−2.0 to 1.3)	0.640	0.09

Mean and standard deviation (SD) values pre- and post-intervention for intermittent pneumatic compression (IPC), along with mean difference (Δ) and its 95% confidence interval (95% CI), *p*-value (paired Student’s *t*-test), and effect size (Cohen’s d). Effect sizes were interpreted as follows: small (≥0.2), medium (≥0.5), large (≥0.8), and very large (>1.2).

**Table 4 sports-13-00304-t004:** Two-way repeated-measures ANOVA (intervention × time) for biochemical (CK) and neuromuscular (Q-IVC, H-IVC, VJ) outcomes in elite under-20 soccer players. Values represent F-statistics with degrees of freedom (df1 = 1, df2 = 19), *p*-values, and partial eta squared (ηp^2^).

Variable	Effect	F	df1	df2	*p*	ηp^2^
CK	Intervention	0.039	1	19	0.846	0.002
	Time	7.168	1	19	0.015 *	0.274
	Intervention × Time	3.618	1	19	0.072	0.160
Q-IVC	Intervention	0.253	1	19	0.621	0.013
	Time	0.293	1	19	0.594	0.015
	Intervention × Time	1.259	1	19	0.276	0.062
H-IVC	Intervention	1.828	1	19	0.192	0.088
	Time	0.108	1	19	0.746	0.005
	Intervention × Time	0.101	1	19	0.754	0.005
VJ	Intervention	1.191	1	19	0.289	0.059
	Time	1.481	1	19	0.238	0.072
	Intervention × Time	0.577	1	19	0.457	0.029

* significant main effect.

## Data Availability

The data that support the findings of this study are available from the corresponding author upon reasonable request.

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
