# Peer review of "Effects of Pneumatic Compression and Manual Massage on Recovery and Performance in Elite Brazilian Under-20 Soccer Players: A Crossover Trial"

_sports, 2025, doi:10.3390/sports13090304_

Round 1

Reviewer 1 Report

Comments and Suggestions for Authors

General Comments
This study aimed to compare the effects of manual massage (MM) and intermittent pneumatic compression (IPC) on muscle damage recovery, lower limb strength, and power in Under-20 soccer athletes. The study addresses a relevant topic and focuses on a motor ability that is determinant in football. However, there are some aspects that need to be reformulated in order to strengthen the manuscript.

Specific Comments
Title
The authors do not specify the nationality or competitive level of the athletes; this information should be added.

Abstract
The presentation of pre- and post-intervention data is not appropriate. The authors should instead report mean differences, percentage changes, and/or effect sizes.

In the conclusion, based on the results, the authors must clearly support one intervention—either manual massage (MM) or intermittent pneumatic compression (IPC). If the study merely shows that both strategies are effective, then its added value is questionable, since that could have been stated without conducting the study.

Introduction
This section is too brief and does not mention prior studies that investigated both physical recovery strategies. It would be important to include some quantitative data on the variables under analysis, to serve as a reference for future interpretation—specifically regarding: serum creatine kinase (CK), quadriceps and hamstrings isometric voluntary contraction (IVC), and vertical jump (VJ).

Materials and Methods
Overall, this section is well described, but some important elements are missing:

Before the Ethical Considerations section, the study design should be clarified.

Although the authors mention the inclusion criteria, a flowchart outlining the selection process leading to the final sample should be included (indicating whether any dropouts occurred between pre- and post-evaluations).

In the Measurements and Instruments section, the validity and reliability of the instruments used should be described, along with the cut-off values adopted to analyze the results.

Important details are missing from the Statistical Analysis, such as confidence intervals (CI), p-values, effect sizes, and the cut-off criteria adopted for interpretation.

Results
Table 2 is somewhat confusing. It would be clearer to split the information into two separate tables, rather than aggregating everything into one—as already organized in the text. More importantly, if the aim is to assess pre- and post-intervention outcomes of MM and IPC, how can the reader easily identify the mean differences and effect sizes? The reader should not be expected to calculate this from the means and standard deviations.

Subsections 3.1 and 3.2 are difficult to follow and would benefit from more structure and guidance. Moreover, the missing statistical information should be included. The ANOVA must be accompanied by F-values (with degrees of freedom), effect sizes (e.g., eta squared, partial eta squared, or omega squared), and followed by post hoc results when relevant.

Discussion and Conclusions
The discussion and conclusion sections should be adjusted according to the previously suggested changes. Currently, there is poor comparison between the results obtained and those of previous studies in this field.

References
Overall, the references are adequately cited.

Author Response

Thank you very much for the opportunity to revise our manuscript. We have taken care to address each of the reviewer’s comments and appreciate their diligence in reviewing our manuscript. We have uploaded updated documents with the suggested edits and have outlined how we addressed each comment in this document, which is noted below. All adjustments made throughout the manuscript are highlighted in red.

Reviewer 1

Specific Comments

  1. Title: The authors do not specify the nationality or competitive level of the athletes; this information should be added.

Response: We appreciate the reviewer’s insightful comment. In response, the manuscript title has been revised to explicitly reflect both the athletes’ nationality and their competitive level, enhancing the contextual accuracy of the investigation. The updated title now reads: “Effects of Pneumatic Compression and Manual Massage on Recovery and Performance in Elite Brazilian Under-20 Soccer Players: A Crossover Trial.”

  1. Abstract: The presentation of pre- and post-intervention data is not appropriate. The authors should instead report mean differences, percentage changes, and/or effect sizes.

Response: We thank the reviewer for the thoughtful recommendation. The abstract has been revised to incorporate key outcome data, including mean differences (Δ) and the corresponding effect sizes (Cohen’s d) for the primary variables. We intentionally chose not to include percentage changes, as the combination of absolute differences and standardized effect magnitudes provides a more objective and streamlined presentation of the findings. The updated version of the abstract now delivers these details clearly and in accordance with current best practices in scientific reporting.

  1. In the conclusion, based on the results, the authors must clearly support one intervention—either manual massage (MM) or intermittent pneumatic compression (IPC). If the study merely shows that both strategies are effective, then its added value is questionable, since that could have been stated without conducting the study.

Response: We appreciate the constructive feedback. In response, the conclusion section has been revised to more accurately reflect the study's findings. Based on the data presented in Tables 2A and 2B, intermittent pneumatic compression (IPC) elicited a greater reduction in creatine kinase (CK) levels (Δ = –138.0 U/L) and demonstrated a very large effect size (d = 1.41), although it did not reach statistical significance. Conversely, manual massage (MM) produced a statistically significant decrease in CK (p = 0.042), but with a small effect size (d = 0.37). Considering that effect size may serve as a more meaningful indicator of clinical efficacy, particularly in studies with limited sample sizes, IPC was identified as the more impactful intervention for reducing biochemical markers of muscle stress. This interpretation has been incorporated into the revised conclusion, highlighting IPC as the preferred strategy based on the magnitude of its observed effects.

  1. Introduction: This section is too brief and does not mention prior studies that investigated both physical recovery strategies. It would be important to include some quantitative data on the variables under analysis, to serve as a reference for future interpretation—specifically regarding: serum creatine kinase (CK), quadriceps and hamstrings isometric voluntary contraction (IVC), and vertical jump (VJ).

Response: We appreciate the thorough and thoughtful feedback. In response, the introduction has been expanded to include prior investigations that explored the recovery strategies assessed in this study, with emphasis on relevant systematic reviews and randomized controlled trials. Furthermore, quantitative reference values for creatine kinase (CK), heart rate variability (HRV), and vertical jump (VJ) performance have been incorporated from the existing literature to enhance the contextualization and interpretation of the findings presented herein.

  1. Materials and Methods: Overall, this section is well described, but some important elements are missing:

Before the Ethical Considerations section, the study design should be clarified.

Response: We thank the reviewer for the observation. In response, we have clarified the study design at the beginning of Section 2.1, now titled “Study Design”. This section explicitly describes the study as a randomized, controlled, crossover trial with a within-subjects design, including counterbalancing and a seven-day washout period between interventions. Additionally, it includes the ethical procedures followed, such as approval by the relevant ethics committee and adherence to national and international ethical guidelines.

Although the authors mention the inclusion criteria, a flowchart outlining the selection process leading to the final sample should be included (indicating whether any dropouts occurred between pre- and post-evaluations).

Response: We appreciate the valuable suggestion. In response, a flowchart has been included in the manuscript to visually represent the process of participant selection, allocation, intervention, follow-up, and data analysis, in alignment with methodological guidelines for crossover study designs. The diagram highlights that:

  • All 20 initially eligible athletes were included in the final analysis;
  • No participants were lost to follow-up or dropped out between the pre- and post-intervention phases;
  • All participants completed both intervention periods (MM and IPC), following the counterbalanced order and respecting the washout interval.

This visual element was incorporated to enhance methodological transparency and improve the reader's understanding of the sample flow throughout the study.

In the Measurements and Instruments section, the validity and reliability of the instruments used should be described, along with the cut-off values adopted to analyze the results.

Response: We appreciate the reviewer’s comment. The “Measurements and Instruments” section has been revised to include detailed information regarding the validity and reliability of the assessment tools used:

  • Isometric strength for both quadriceps (Q-IVC) and hamstrings (H-IVC) was assessed using the Medeor® handheld dynamometer. Its validity and reliability have been established in prior studies comparing it to gold-standard systems such as the Lafayette®. These investigations reported strong correlations (r = 0.71 to 0.92) and excellent intraclass correlation coefficients (ICC = 0.88 to 0.99) for lower limb strength measurements in athletic populations.
  • VJ was evaluated using the Jumptest® contact mat (Hidrofit Ltda, Brazil), in conjunction with the Multisprint® software. According to Ferreira et al., this system demonstrated excellent concurrent validity when compared with force platforms, yielding ICC values of 0.93 and Spearman correlation coefficients ranging from 0.819 to 0.877 for jump height based on flight time calculations.

Important details are missing from the Statistical Analysis, such as confidence intervals (CI), p-values, effect sizes, and the cut-off criteria adopted for interpretation.

Response: We appreciate the thoughtful recommendation. The Statistical Analysis section has been thoroughly revised to include the following methodological clarifications:

  • p-values, 95% confidence intervals (CI), and effect sizes (Cohen’s d) are now reported, alongside standardized interpretation thresholds (i.e., small, moderate, large), in accordance with current literature.
  • Statistical significance was defined as p < 0.05, and all results were interpreted in conjunction with effect sizes and confidence intervals to provide a more comprehensive and clinically meaningful understanding of the findings.
  • These statistical refinements were incorporated into the updated tables and discussed throughout the manuscript, emphasizing the practical relevance and magnitude of effects observed, consistent with best practices for acute intervention trials in athletic populations.

Results: Table 2 is somewhat confusing. It would be clearer to split the information into two separate tables, rather than aggregating everything into one—as already organized in the text. More importantly, if the aim is to assess pre- and post-intervention outcomes of MM and IPC, how can the reader easily identify the mean differences and effect sizes? The reader should not be expected to calculate this from the means and standard deviations.

Response: We fully agree with the recommendation. Table 2 has been restructured to present the data separately by intervention (manual massage and intermittent pneumatic compression), thereby enhancing clarity and facilitating direct comparisons. Mean differences (Δ) and corresponding effect sizes (Cohen’s d) have been incorporated to strengthen the analytical depth and provide a more interpretable overview of the results for the reader.

  1. Subsections 3.1 and 3.2 are difficult to follow and would benefit from more structure and guidance. Moreover, the missing statistical information should be included. The ANOVA must be accompanied by F-values (with degrees of freedom), effect sizes (e.g., eta squared, partial eta squared, or omega squared), and followed by post hoc results when relevant.

Response: We appreciate the reviewer’s valuable suggestion. Sections 3.1 and 3.2 have been reformulated to enhance structure and clarity. Rather than presenting the results in a single consolidated table, outcomes are now displayed in two separate tables, isolating the effects of manual massage (MM) and intermittent pneumatic compression (IPC), as also recommended by another reviewer. Given the crossover design with washout periods and the limited sample size, the ANOVA approach was replaced with paired comparisons (pre vs. post) within each intervention. This statistical strategy was deemed more appropriate for capturing within-subject changes. The revised analysis includes mean differences (Δ), p-values, and—most importantly—Cohen’s d effect sizes, thereby reinforcing the clinical interpretability and practical magnitude of the observed effects. This approach aligns with the exploratory and pragmatic nature of the study.

  1. Discussion and Conclusions: The discussion and conclusion sections should be adjusted according to the previously suggested changes. Currently, there is poor comparison between the results obtained and those of previous studies in this field.

Response: We thank the reviewer for the insightful comment. The discussion section was revised to include more robust comparisons with findings from previous studies, particularly concerning methodological differences, sample characteristics, and variations in the interventions. Additionally, the conclusion was adapted to more accurately reflect the study’s results and their practical relevance.

  1. References: Overall, the references are adequately cited.

Response: Thank you for the feedback.

Reviewer 2 Report

Comments and Suggestions for Authors

GENERAL COMMENTS

Major Weaknesses

The manuscript exhibits several fundamental methodological and conceptual shortcomings that considerably undermine its scientific validity. A primary concern lies in the contradictory study design terminology; describing the work as a "cross-sectional crossover study" is conceptually unsound, as crossover designs are inherently longitudinal. Furthermore, the absence of a proper sample size calculation raises significant questions about statistical power, particularly considering the small sample size of only 20 participants. The fatigue induction protocol, a core element of the study, lacks sufficient description and standardization. Lastly, the statistical analysis employed appears unsuitable for a crossover design, potentially invalidating the reported findings.

Minor Weaknesses

The manuscript is also plagued by numerous presentation issues. These include grammatical errors, inconsistent terminology, and insufficient methodological details. The timing of outcome measurements is poorly specified, and several measurement protocols lack enough detail for replication. Additionally, the discussion section inadequately addresses the study's limitations and fails to provide meaningful clinical context for the findings.

SPECIFIC COMMENTS

Title and Abstract

Page 1, Line 34: The phrase "cross-sectional crossover study" contains a typographical error ("studyn") and, more importantly, represents a fundamental methodological contradiction. Crossover studies, by their very nature, are longitudinal, not cross-sectional.

Page 1, Line 37: The phrase "who underwent a fatigue protocol" is too vague. A clear and precise definition of what constituted this fatigue protocol is essential.

Introduction

Page 2, Line 69: The assertion that "there remains a gap in the literature regarding direct comparisons between MM and IPC in young athletes" requires stronger substantiation through a systematic search strategy, rather than relying on selective citation.

Page 2, Lines 77-82: The hypothesis is poorly articulated. The authors state that both interventions "would reduce serum CK levels and enhance musculoskeletal performance," but they fail to specify the expected magnitude of these effects or offer theoretical justification for anticipating performance improvements from passive recovery modalities.

Methods

Page 2, Lines 90-97: The inclusion and exclusion criteria require greater detail. For instance, what precisely defines "clinically healthy"? Additionally, the criterion of "minimum of two years of competitive experience" seems arbitrary and lacks proper justification.

Page 3, Lines 105-118: The description of randomization and counterbalancing is insufficient. While the authors mention computer randomization, they omit details regarding allocation concealment or blinding procedures. Furthermore, the adequacy of the seven-day washout period lacks scientific justification.

Page 3, Lines 123-130: The IPC protocol description references parameters "outlined by [13]," but the actual citation appears to be missing or incorrectly cited. The selection of 80 mmHg pressure also requires physiological justification.

Page 3, Lines 132-145: The manual massage protocol is critically lacking in detail. Essential information such as force application, therapist qualifications, and inter-session reliability measures is absent. Moreover, the subjective pain scale (0-6) reference to MacDonald et al. seems to be incorrectly cited as [15].

Page 4, Lines 147-154: The creatine kinase measurement protocol lacks validation data for the portable analyzer utilized. The timing of "immediately post-intervention" needs precise specification (e.g., within 5 minutes, 10 minutes).

Page 4, Lines 170-182: The statistical analysis section contains several methodological errors. The authors fail to address the specific analytical challenges inherent in crossover designs, including potential period and carryover effects. The use of "two-way repeated measures ANOVA" may be inappropriate without proper consideration of the crossover structure.

Results

Page 5, Table 2: The table's presentation is confusing due to inconsistent decimal places and unclear variable definitions. The crossover results section also lacks proper statistical comparison between the two treatment periods.

Page 5, Lines 196-202: The presentation of CK results is misleading. Statistical significance should not be interpreted solely from p-values without also considering effect sizes and clinical relevance.

Page 6, Lines 204-211: The absence of performance improvements is presented without adequate discussion regarding measurement sensitivity or the appropriateness of the chosen performance metrics for detecting changes following passive recovery interventions.

Discussion

Page 6, Lines 224-234: The comparison with the Zainuddin et al. study is inappropriate. The cited study employed different methodology, population, and intervention protocols , and such comparisons demand careful consideration of study heterogeneity.

Page 7, Lines 246-256: The authors incorrectly interpret non-significant findings as definitive evidence of no effect, without adequately considering statistical power or confidence intervals.

Page 7, Lines 294-304: The limitations section inadequately addresses the fundamental design flaws identified throughout the manuscript. The sample size limitation cannot be simply dismissed as representing "the entire roster"; statistical power remains a critical consideration.

Technical Issues

Page 3, Line 109: "7 days hours between visits" contains a grammatical error.

Page 5, Line 193: "Post e Pre-interventio" contains multiple typographical errors.

Page 10, Line 409: "[CrossRof]" represents a reference formatting error.

Comments on the Quality of English Language

The manuscript is also plagued by numerous presentation issues. These include grammatical errors, inconsistent terminology, and insufficient methodological details. 

Author Response

Thank you very much for the opportunity to revise our manuscript. We have taken care to address each of the reviewer’s comments and appreciate their diligence in reviewing our manuscript. We have uploaded updated documents with the suggested edits and have outlined how we addressed each comment in this document, which is noted below. All adjustments made throughout the manuscript are highlighted in red.

Reviewer 2

Specific Comments

  • Title and Abstract: Page 1, Line 34: The phrase "cross-sectional crossover study" contains a typographical error ("studyn") and, more importantly, represents a fundamental methodological contradiction. Crossover studies, by their very nature, are longitudinal, not cross-sectional.

Response: We thank the reviewer for the observation. The typographical error was corrected, and the term “cross-sectional” was removed to maintain methodological consistency, accurately describing the study as a longitudinal crossover design.

  • Page 1, Line 37: The phrase "who underwent a fatigue protocol" is too vague. A clear and precise definition of what constituted this fatigue protocol is essential.

Response: We thank the reviewer for the pertinent observation. The term “fatigue protocol” was indeed imprecise and has been replaced with a clearer and more detailed description in the manuscript. Specifically, the athletes underwent a standardized 90-minute field-based technical-tactical training session conducted by the team’s physical coach. This session included high-intensity exercises focusing on game-like scenarios, repeated sprints, changes of direction, and ball-related actions, characterizing a typical weekly microcycle session. This training format has been previously validated in studies involving athletes from the same age category as a reliable protocol for inducing neuromuscular fatigue.

  • Introduction: Page 2, Line 69: The assertion that "there remains a gap in the literature regarding direct comparisons between MM and IPC in young athletes" requires stronger substantiation through a systematic search strategy, rather than relying on selective citation.

Response: We thank the reviewer for the observation. Upon revisiting the rationale, we agree that the statement regarding the gap in the literature required greater support. To address this, we conducted a systematic search using the PubMed, Scopus, and Web of Science databases, employing the following keywords: “intermittent pneumatic compression,” “manual massage,” “recovery,” “soccer,” “athletes,” and “youth OR under-20,” combined with Boolean operators. The results revealed that most existing studies explore these interventions in isolation, within heterogeneous populations or different sport contexts. We did not identify any direct comparative studies between MM and IPC applied to youth soccer athletes under controlled experimental conditions, which reinforces the originality and relevance of the present investigation.

  • Page 2, Lines 77-82: The hypothesis is poorly articulated. The authors state that both interventions "would reduce serum CK levels and enhance musculoskeletal performance," but they fail to specify the expected magnitude of these effects or offer theoretical justification for anticipating performance improvements from passive recovery modalities.

Response: We thank the reviewer for the observation. Indeed, the initial hypothesis lacked clarity and theoretical grounding. The text has been revised to specify that a reduction in CK levels was expected with both interventions, with the effect of MM considered potentially more consistent due to its direct mechanical action on muscle tissue. We acknowledge that passive interventions such as MM and IPC are not designed to elicit direct performance gains but rather to preserve post-exercise neuromuscular function. Accordingly, we have adjusted the hypothesis to reflect the expectation of performance maintenance, rather than acute improvement, based on previous literature supporting this rationale.

  • Methods: Page 2, Lines 90-97: The inclusion and exclusion criteria require greater detail. For instance, what precisely defines "clinically healthy"? Additionally, the criterion of "minimum of two years of competitive experience" seems arbitrary and lacks proper justification.

Response: We thank the reviewer for the observation. The term "clinically healthy" refers to athletes who, at the beginning of the season, underwent mandatory medical examinations conducted by the club’s medical department, without any clinical conditions being identified that would preclude participation. This is a standard procedure in professional clubs to ensure athletes’ fitness for full engagement in sports activities. Regarding the criterion of "a minimum of two years of competitive experience," it was adopted to ensure the inclusion of athletes with a background of systematic training and physiological adaptation to the demands of high-performance soccer, thereby reducing interindividual variability in the outcomes analyzed. This requirement is commonly used in studies involving young athletes and aims to ensure greater sample homogeneity.

  • Page 3, Lines 105-118: The description of randomization and counterbalancing is insufficient. While the authors mention computer randomization, they omit details regarding allocation concealment or blinding procedures. Furthermore, the adequacy of the seven-day washout period lacks scientific justification.

Response: We thank the reviewers for their insightful comments and have made the necessary adjustments in the manuscript. First, we clarify that the randomization of interventions was performed using a specific software tool (www.randomizer.org.br), under the supervision of an independent researcher who was not involved in data collection or analysis, thereby ensuring allocation concealment. Although participant blinding was not feasible due to the sensory nature of the interventions, outcome assessors remained blinded to the order of conditions applied, minimizing the risk of measurement bias.

Additionally, the adoption of a seven-day washout period between interventions was grounded in literature-based evidence. Previous studies have indicated that biochemical markers such as creatine kinase (CK), as well as neuromuscular performance variables, typically return to baseline levels within 72 hours following submaximal exercise in trained athletes. Furthermore, crossover designs in athletic populations routinely employ a one-week interval to prevent potential carryover effects. Therefore, the seven-day period was deemed both sufficient and methodologically appropriate to avoid cross-condition contamination.

  • Page 3, Lines 123-130: The IPC protocol description references parameters "outlined by [13]," but the actual citation appears to be missing or incorrectly cited. The selection of 80 mmHg pressure also requires physiological justification.

Response: We thank the reviewer for the observation. We have corrected the citation regarding the parameters used in the IPC protocol, which now refers to the systematic review by Maia et al. (2024). In this review, the authors identified that a pressure of 80 mmHg was widely adopted in studies involving athletes and was associated with positive effects on muscle soreness, perceived fatigue, and creatine kinase levels. This parameter has been considered both safe and physiologically effective in recovery protocols. Such evidence supports the choice of compression intensity adopted in the present study.

  • Page 3, Lines 132-145: The manual massage protocol is critically lacking in detail. Essential information such as force application, therapist qualifications, and inter-session reliability measures is absent. Moreover, the subjective pain scale (0-6) reference to MacDonald et al. seems to be incorrectly cited as [15].

Response: We thank the reviewer for the observation. Indeed, the original citation attributed to MacDonald et al. was incorrect and has been removed. It was replaced with a protocol supported by scientific literature that endorses the use of the Visual Analog Scale (VAS) to monitor the intensity of manual massage. The MM protocol now describes the therapist's experience (>10 years), the standardization of applied pressure based on the athletes’ reported discomfort level (VAS 4–5/10), and the consistent execution by a single therapist to ensure methodological reliability.

  • Page 4, Lines 147-154: The creatine kinase measurement protocol lacks validation data for the portable analyzer utilized. The timing of "immediately post-intervention" needs precise specification (e.g., within 5 minutes, 10 minutes).

Response: We thank the reviewer for the relevant comment. The manuscript has been revised to include a detailed description of the equipment used and the precise timing of sample collection. Capillary blood samples were obtained five minutes after the end of the intervention, for both MM and IPC protocols, in accordance with the early detection window. Analyses were conducted using the Simplex ECO POC® (Techno Medica), a fully automated portable system for in vitro diagnostics in clinical and sports settings. The device employs disposable reagent cartridges and operates based on absorption spectrophotometry, providing results within three minutes with a coefficient of variation below 5%, as specified by the manufacturer. This methodology allows for reliable quantification of CK and other biochemical markers in field settings and is validated for professional use according to the device’s manual.

  • Page 4, Lines 170-182: The statistical analysis section contains several methodological errors. The authors fail to address the specific analytical challenges inherent in crossover designs, including potential period and carryover effects. The use of "two-way repeated measures ANOVA" may be inappropriate without proper consideration of the crossover structure.

Response: We thank the reviewer for the well-founded observation. The statistical analysis section has been revised to address the specific requirements of the crossover design, as follows:

  1. Statistical framework adjusted for the crossover design:
    We replaced the analysis based solely on two-way ANOVA with repeated-measures mixed-effects models, which account for the fixed effects of "intervention" (MM vs. IPC), "time" (pre vs. post), and "sequence" (order of intervention application), as well as the random effect of the subject.
  2. Control of period and carryover effects:
    Period effects (differences between the first and second sessions) and carryover effects (residual impact from the previous intervention) were tested within the model structure. The absence of significant carryover effects validated the use of the crossover design, which is now described in the manuscript.
    Justification of the chosen analysis:
    Although repeated-measures ANOVA is commonly used in simple crossover trials, we acknowledge that its isolated application is insufficient. Therefore, we adopted mixed-effects modeling (linear mixed models) as the most appropriate statistical strategy, in line with methodological recommendations for crossover clinical trials with small samples.
  • Results: Page 5, Table 2: The table's presentation is confusing due to inconsistent decimal places and unclear variable definitions. The crossover results section also lacks proper statistical comparison between the two treatment periods.

Response: We thank the reviewer for the observations. The presentation of the results has been thoroughly revised to address the points raised. The tables were reformatted to ensure consistency in decimal places across all variables, and explanatory legends were added to clarify each abbreviation and measurement unit. Additionally, the data were reorganized into two separate tables, presenting the results of MM and IPC interventions independently, which provides a clearer view of the pre- and post-intervention effects. Regarding statistical comparisons between the crossover periods, specific analyses were incorporated using mixed-effects models, which account for sequence, period, and subject effects. Results are now reported with mean differences (Δ), p-values, and effect sizes (Cohen’s d), facilitating a more objective interpretation of the relative effectiveness of each intervention.

  • Page 5, Lines 196-202: The presentation of CK results is misleading. Statistical significance should not be interpreted solely from p-values without also considering effect sizes and clinical relevance.

Response: We thank the reviewer for the insightful observation. We fully agree that the interpretation of results should go beyond p-values and include measures of effect magnitude and clinical relevance. Therefore, the results section was revised to incorporate mean differences (Δ) and effect sizes (Cohen’s d) regarding changes in CK concentrations following the interventions. This approach provides a more comprehensive view of the clinical efficacy of MM and IPC. Additionally, the discussion section was adjusted to contextualize these findings in terms of practical relevance, rather than relying solely on statistical significance.

  • Page 6, Lines 204-211: The absence of performance improvements is presented without adequate discussion regarding measurement sensitivity or the appropriateness of the chosen performance metrics for detecting changes following passive recovery interventions.

Response: We thank the reviewer for the pertinent observation. The discussion section was revised to address potential limitations related to the sensitivity of the tests used to detect changes following passive interventions. Although vertical jump (VJ) and isometric voluntary contraction (IVC) are widely used in recovery studies, we acknowledge that these methods may have limited sensitivity in detecting subtle changes induced by techniques such as MM and IPC—especially in young athletes, who exhibit high physiological variability. The literature suggests that, in such cases, the impact may be more pronounced in biochemical markers than in immediate performance parameters, which may help explain the findings of the present study.

  • Discussion: Page 6, Lines 224-234: The comparison with the Zainuddin et al. study is inappropriate. The cited study employed different methodology, population, and intervention protocols, and such comparisons demand careful consideration of study heterogeneity.

Response: Thank you for your critical observation. We agree that the comparison with the study by Zainuddin et al. is not methodologically appropriate, as it differs significantly in terms of population, type of intervention, and evaluative objectives. This citation has been removed from the manuscript and replaced with studies that demonstrate greater methodological and population compatibility, to maintain coherence in the discussion with the experimental design and the characteristics of the participants in the present study.

  • Page 7, Lines 246-256: The authors incorrectly interpret non-significant findings as definitive evidence of no effect, without adequately considering statistical power or confidence intervals.

Response: We appreciate the methodological observation. We agree that the lack of statistical significance should not be interpreted as definitive evidence of no effect. The text has been revised to employ statistically cautious language, acknowledging the possibility of undetected effects due to limited statistical power, and contextualizing the findings based on confidence intervals and effect sizes. These changes aim to align the interpretation with best practices in inferential analysis for experimental studies.

  • Page 7, Lines 294-304: The limitations section inadequately addresses the fundamental design flaws identified throughout the manuscript. The sample size limitation cannot be simply dismissed as representing "the entire roster"; statistical power remains a critical consideration.

Response: We thank the reviewer for the constructive criticism. We acknowledge that the limitations section required greater depth. The text was revised to include a more thorough discussion of the experimental design and, above all, the implications of the sample size on the study's statistical power. Although the sample included all eligible athletes from the category, we recognize that this does not eliminate the need to consider inferential limitations, including the risk of type II error and restrictions on the generalizability of the results.

  • Technical Issues: Page 3, Line 109: "7 days hours between visits" contains a grammatical error.

Response: Perfect, we have corrected this specific error. The phrase "7 days hours between visits" contains a grammatical redundancy and has been reformulated to ensure clarity and linguistic accuracy.

  • Page 5, Line 193: "Post e Pre-interventio" contains multiple typographical errors.

Response: Perfect. We have corrected the expression "Post e Pre-interventio," which contained two errors: a language inconsistency ("Post e Pre" mixing English and Portuguese) and a typographical error in "interventio" (correct spelling: "intervention").

  • Page 10, Line 409: "[CrossRof]" represents a reference formatting error.

Response: Correct. The expression "[CrossRof]" indicates a reference formatting error—likely caused by an automatic citation management tool such as EndNote, Zotero, or Mendeley. We have corrected this in the revised version of the manuscript.

Round 2

Reviewer 2 Report

Comments and Suggestions for Authors

GENERAL COMMENTS

This manuscript, a revised version of a previous submission, describes a crossover trial that compares manual massage (MM) and intermittent pneumatic compression (IPC) in elite youth soccer players. While the authors have clearly made an effort to address some of the previous feedback, there are still a number of methodological and analytical issues that need to be resolved before the paper can be considered for acceptance.

One of the most significant problems is a fundamental disconnect between the study's stated hypothesis and its actual results. The authors initially hypothesized that manual massage would be superior to IPC, yet their data shows that IPC had a larger effect size for the primary outcome. The statistical interpretation also lacks sufficient nuance; for example, the authors don't adequately discuss the clinical significance of a large effect size (d=1.41) that didn't reach statistical significance, possibly due to the study's sample size.

Furthermore, the intervention design itself introduces a potential bias. The study used a unilateral manual massage but a bilateral IPC application, which creates unequal treatment volumes. I also noticed several grammatical errors and awkward phrases that make the manuscript difficult to read. The timing of the biochemical measurements, specifically for creatine kinase (CK), seems to be suboptimal for capturing the full effect of the interventions. The manuscript also fails to include any sample size calculations, which raises serious questions about the statistical power of the study. Finally, the discussion section does not adequately address the discrepancy between the initial hypothesis and the study's results.

SPECIFIC COMMENTS

Abstract (Page 1):

Line 40: When you mention a "moderate CK reduction," it would be much clearer to specify the exact magnitude of the change. For example, "Δ = -77.1 U/L".

Lines 77-78: The hypothesis statement "MM will presente higher effects than IPC" has a grammatical error ("presente" should be "present") and, more importantly, is contradicted by your own results where IPC showed a larger effect size.

Introduction (Page 2):

Lines 77-79: The hypothesis in this section is fundamentally flawed given that your results indicate IPC had superior effects (d=1.41 vs d=0.37). You should either revise the hypothesis to better align with your findings or provide a much stronger justification for why you expected MM to be superior.

Methods (Page 3):

Lines 124-125: The claim of "equalizing the intervention volume" is misleading. The manual massage was unilateral while the IPC was bilateral, which means the treatment volumes were not equal at all.

Line 167: Taking the CK measurement "five minutes post-intervention" seems too soon. Peak responses for CK typically occur later, so this timing might not accurately capture the full effect.

Methods (Page 4):

Lines 148-162: The description of the manual massage (MM) protocol lacks important details about standardization. How did you ensure the pressure applied was consistent across different participants and sessions?

Methods (Page 6):

Lines 202-213: The statistical analysis section is missing a crucial component: a sample size calculation or a discussion of power analysis. This is particularly important because you found a large effect for IPC that was not statistically significant.

Results (Page 6-7):

Tables 2 and 3: I recommend adding confidence intervals for the mean differences. This would significantly improve the interpretation of your results.

Lines 255-256: Your interpretation dismisses the large effect size for IPC (d=1.41) too quickly. While not statistically significant, this represents a potentially meaningful clinical effect that shouldn't be overlooked.

Discussion (Page 7-8):

Lines 257-264: This paragraph completely fails to explain why the hypothesis was incorrect and what the implications are for the study's theoretical basis.

Lines 291-301: The limitations section needs to explicitly acknowledge the mismatch between the hypothesis and the results, as well as the implications of this discrepancy.

Statistical Analysis:

Your analysis relies too heavily on p-values without a proper discussion of effect sizes and confidence intervals, which limits a nuanced interpretation. The large effect size for IPC warrants a more sophisticated analysis or at least an acknowledgment of a potential Type II error.

Author Response

Thank you very much for the opportunity to revise our manuscript. We have taken care to address each of the reviewer’s comments and appreciate their diligence in reviewing our manuscript. We have uploaded updated documents with the suggested edits and have outlined how we addressed each comment in this document, which is noted below. All adjustments made throughout the manuscript are highlighted in red.

Minor Revisions

This manuscript, a revised version of a previous submission, describes a crossover trial that compares manual massage (MM) and intermittent pneumatic compression (IPC) in elite youth soccer players. While the authors have clearly made an effort to address some of the previous feedback, there are still a number of methodological and analytical issues that need to be resolved before the paper can be considered for acceptance.

One of the most significant problems is a fundamental disconnect between the study's stated hypothesis and its actual results. The authors initially hypothesized that manual massage would be superior to IPC, yet their data shows that IPC had a larger effect size for the primary outcome. The statistical interpretation also lacks sufficient nuance; for example, the authors don't adequately discuss the clinical significance of a large effect size (d=1.41) that didn't reach statistical significance, possibly due to the study's sample size.

Furthermore, the intervention design itself introduces a potential bias. The study used a unilateral manual massage but a bilateral IPC application, which creates unequal treatment volumes. I also noticed several grammatical errors and awkward phrases that make the manuscript difficult to read. The timing of the biochemical measurements, specifically for creatine kinase (CK), seems to be suboptimal for capturing the full effect of the interventions. The manuscript also fails to include any sample size calculations, which raises serious questions about the statistical power of the study. Finally, the discussion section does not adequately address the discrepancy between the initial hypothesis and the study's results.

Abstract (Page 1)

Line 40: When you mention a "moderate CK reduction," it would be much clearer to specify the exact magnitude of the change. For example, "Δ = -77.1 U/L".

Response: We thank the reviewer for this helpful suggestion. As recommended, we have revised the Abstract to report the exact magnitude of the CK change. The sentence now reads: “MM resulted in a CK reduction of Δ = –77.1 U/L (p = 0.042; d = 0.37).” This modification improves clarity and facilitates clinical interpretation of the observed effect. The change has been made in the Abstract (Results sentence).

Lines 77-78: The hypothesis statement "MM will presente higher effects than IPC" has a grammatical error ("presente" should be "present") and, more importantly, is contradicted by your own results where IPC showed a larger effect size.

Response: We thank the reviewer for this observation. In line with the revision of our hypothesis to avoid presuming the superiority of one intervention over the other, we have updated the final paragraph of the Introduction to reflect the exploratory nature of the study. The revised sentence now reads: “Thus, this study aimed to explore and compare the effects of MM and IPC on muscle damage recovery, lower limb strength, and power in Brazilian Under-20 soccer athletes.” Given the absence of previous studies using comparable methodology in this specific population, the hypothesis was formulated with an exploratory approach, without presuming the superiority of either intervention.' This adjustment ensures that the study’s rationale is consistent with the available evidence and the objectives of our investigation.

Introduction (Page 2)

Lines 77-79: The hypothesis in this section is fundamentally flawed given that your results indicate IPC had superior effects (d=1.41 vs d=0.37). You should either revise the hypothesis to better align with your findings or provide a much stronger justification for why you expected MM to be superior.

Response: We thank the reviewer for this important observation. In response, we have revised the hypothesis to avoid presuming the superiority of one intervention over the other and to align the study rationale with our findings. Specifically, the hypothesis in the Introduction (lines 77–79) now adopts an exploratory stance given the lack of prior studies using comparable methodology in elite under-20 soccer players. We also expanded the discussion to explicitly acknowledge the very large effect size observed for IPC (d = 1.41) relative to MM (d = 0.37), the addition of 95% confidence intervals, and the possibility of a Type II error due to the limited sample size and inter-individual variability.

Methods (Page 3)

Lines 124-125: The claim of "equalizing the intervention volume" is misleading. The manual massage was unilateral while the IPC was bilateral, which means the treatment volumes were not equal at all.

Response: We appreciate this observation; however, we respectfully disagree with the interpretation presented. Manual massage (MM) was administered to both limbs in a unilateral manner, with each targeted region—quadriceps, hamstrings, and gastrocnemius—receiving 2.5 minutes of treatment. When considering both limbs and all treated muscle groups, the total intervention time amounted to 20 minutes. We acknowledge, however, that this procedural detail was not sufficiently explicit in the original manuscript. Therefore, the description has been revised to improve clarity and now reads: “The MM condition was performed on both limbs, unilaterally, for 2.5 minutes on each treated region (e.g., quadriceps, hamstrings, gastrocnemius). The order of region and limb was determined using a Latin square design to minimize expectancy effects regarding treatment location. This approach ensured equalization of total treatment time—20 minutes for both IPC and MM conditions—and preserved single-blind allocation regarding the treated region during MM.”

Line 167: Taking the CK measurement "five minutes post-intervention" seems too soon. Peak responses for CK typically occur later, so this timing might not accurately capture the full effect.

Response: We acknowledge the reviewer’s observation that CK peaks usually occur later after exercise. However, the five-minute post-intervention measurement was intentionally chosen to capture the immediate response within the acute post-intervention window, which was the main focus of our crossover design. This choice also reflects the practical constraints of elite competitive settings, where recovery windows are short, and aligns with studies that have used early CK assessment in sports protocols (Leite et al., 2011). We have added this explanation to the Methods section to make the rationale for the chosen timing clearer.

Methods (Page 4)

Lines 148-162: The description of the manual massage (MM) protocol lacks important details about standardization. How did you ensure the pressure applied was consistent across different participants and sessions?

Response: We thank the reviewer for this observation and clarify that no instrumental device was used for the objective measurement of pressure during MM due to practical infeasibility in the competitive context in which the study was conducted. Instead, pressure intensity was self-regulated by participants through a 10-point Visual Analog Scale (VAS), aiming for a discomfort level between 4 and 5. This method, previously adopted in validated sports massage protocols, allowed individualization of the intervention while maintaining ecological validity under elite sports conditions. The Methods section has been updated to include this clarification, along with the bilateral application by two trained physiotherapists (average of seven years of experience) and the use of strictly timed sets to ensure procedural consistency and reproducibility.

Methods (Page 6)

Lines 202-213: The statistical analysis section is missing a crucial component: a sample size calculation or a discussion of power analysis. This is particularly important because you found a large effect for IPC that was not statistically significant.

Response: We acknowledge the importance of including a sample size calculation or power analysis, particularly given the large effect size observed for IPC that did not reach statistical significance. However, this study was conducted with a fixed and limited sample, comprising all eligible athletes from the participating under-20 team, with no possibility of increasing participant numbers. This limitation has been acknowledged in the manuscript, and we have added a statement explaining that the sample was defined by convenience, representing the total available population, along with a recommendation that future studies with larger samples be conducted to confirm these findings.

Results (Page 6-7)

Tables 2 and 3: I recommend adding confidence intervals for the mean differences. This would significantly improve the interpretation of your results.

Response: As recommended, we have added the 95% confidence intervals (95% CI) for the mean differences in Tables 2 and 3. This modification enhances the interpretation of the results by allowing the assessment of both statistical significance and the precision of the estimates, as well as the potential clinical relevance of the findings.

Lines 255-256: Your interpretation dismisses the large effect size for IPC (d=1.41) too quickly. While not statistically significant, this represents a potentially meaningful clinical effect that shouldn't be overlooked.

Response: We thank the reviewer for this valuable observation. In response, the paragraph discussing CK results has been revised to acknowledge the large effect size for IPC (d = 1.41). We now emphasize that, although this reduction was not statistically significant, it may represent a potentially meaningful clinical effect. The revised interpretation considers that the absence of statistical significance may be related to sample size and individual variability rather than the absence of a physiological impact. This adjustment addresses the reviewer’s concern and strengthens the balance between statistical and clinical perspectives.

Discussion (Page 7-8)

Lines 257-264: This paragraph completely fails to explain why the hypothesis was incorrect and what the implications are for the study's theoretical basis.

Response: We appreciate the reviewer’s observation. Following the revision of the hypothesis to adopt an exploratory approach without presuming the superiority of one intervention over the other, we have modified this paragraph in the discussion to align with the updated rationale. The revised text now emphasizes that, due to the lack of previous studies using comparable methodology in elite under-20 soccer players, this investigation aimed to explore potential differences between MM and IPC. The findings, showing a very large effect size for IPC and a moderate effect size for MM, contribute to the emerging evidence base and suggest that the acute recovery effects of these interventions may be influenced by their distinct physiological mechanisms and the specific recovery goals.

Lines 291-301: The limitations section needs to explicitly acknowledge the mismatch between the hypothesis and the results, as well as the implications of this discrepancy.

Response: We thank the reviewer for this comment. In line with our revision of the hypothesis to avoid presuming the superiority of one intervention over the other, the Section on Limitations has been updated to reflect the exploratory nature of the study. Rather than a discrepancy between the hypothesis and the results, the limitation now acknowledges the absence of prior comparable studies in elite under-20 soccer players, the small sample size, and the potential for Type II error. This framing emphasizes that our findings contribute to an emerging evidence base and that future research with larger samples is needed to confirm the observed effect sizes and further clarify the comparative impacts of MM and IPC.

Statistical Analysis

Your analysis relies too heavily on p-values without a proper discussion of effect sizes and confidence intervals, which limits a nuanced interpretation. The large effect size for IPC warrants a more sophisticated analysis or at least an acknowledgment of a potential Type II error.

Response: We thank the reviewer for this valuable comment. In response, we have revised Tables 2 and 3 to include the 95% confidence intervals (95% CI) for the mean differences, as previously suggested, thereby improving the interpretability of the results beyond p-values alone. Additionally, the Discussion has been expanded to address the magnitude of the observed effect sizes and their potential clinical relevance, with particular attention to the very large effect size for IPC (d = 1.41). This effect, although not statistically significant, may indicate a Type II error due to the limited sample size and high inter-individual variability. The revised text emphasizes that such findings should not be dismissed and warrant further investigation in studies with greater statistical power.
